# Prevalence and incidence of Coats disease in a large claims database

**Nobuhisa Ochiai[1], Kenji Fujimoto[2], Kazuma Oku[1], Hiroyuki Kondo[1]***

**1** Department of Ophthalmology, Faculty of Medicine, University of Occupational and Environmental Health, Kitakyushu, Fukuoka, Japan, **2** Occupational Health Data Science Center, University of Occupational and Environmental Health, Kitakyushu, Fukuoka, Japan

* kondohi@med.uoeh-u.ac.jp

## Abstract

### Purpose

To determine the prevalence and incidence of Coats disease in the Japan Medical Data Center (JMDC) claims database.

### Methods

Data on patients with Coats disease were collected from the JMDC claims database that were collected over a five-year period between 2019–2023. The patients with were grouped by their sex and age. The prevalence and incidence were then calculated relative to all of the members in the JMDC database. The treatment rate was calculated as the ratio of patients receiving retinal photocoagulation or vitrectomy to the total number of patients over the five-year period.

### Results

The age-adjusted prevalence of Coats disease in the JMDC database was 1.7/100,000 with 2.4 males to 0.9 females for the five-year period. The age-adjusted prevalence in children <20-years-of-age with Coats disease was 3.6/100,000 with 5.7 boys to 1.5 girls, while that of adults age ≥ 20 years was 1.3/100,000 with 1.7 males to 0.7 females. The age-adjusted incidence of Coats disease was 0.7/100,000 individuals with 1.0 males to 0.4 females, and for different ages, the age-adjusted incidence was 1.4/100,000 children (2.1 boys to 0.6 girls), and 0.6/100,000 adults (0.8 men to 0.3 women). The treatment rate was 21.9% for all Coats individuals with children having a significantly higher treatment rate of 28.0% than adults of 18.2%.

### Conclusion

This is the first study that determined the prevalence and incidence of Coats disease in the JMDC database. The prevalence and incidence in children were approximately

**Data availability statement:** The dataset used in this study was extracted from the JMDC claims Database, which is owned by JMDC Inc. Other qualified researchers can access the same dataset by paying the required usage fee and entering into a data use agreement with JMDC Inc. For inquiries regarding data access, please contact JMDC Inc. via their website: https://www.jmdc.co.jp/en/inquiry/. Therefore, the authors did not have any special access privileges that others would not have.

**Funding:** NO received JSPS KAKENHI (grant number 24K12756). The Japan Society for the Promotion of Science (JSPS). http://www.jsps.go.jp/english/e-grants/index.html. The funders had no role in study design, data collection and analysis, decision to publish, or preparation of the manuscript.

**Competing interests:** The authors have declared that no competing interests exist.

**Abbreviations:** JMDC, Japan medical data center; ICD-10, international classification of diseases, 10th revision; 95% CI; 95% confidence intervals; UK, United Kingdom; BOSU, British ophthalmological surveillance unit.

double that of adults. The proportion of females was higher among adults than among children.

---

## Introduction

Coats disease is a disorder of the eyes of unknown cause. It is characterized by the dilation of the retinal vessels and exudative lesions in the retina. It was first identified by Coats in 1908 [1], and Shields et al. classified Coats disease into 5 clinical stages in 2001 [2]. In the early stages, capillary telangiectasia is observed which progresses to exudative retinal detachments and total retinal detachments. Subsequently, neovascular glaucoma develops leading to blindness in the terminal stages of the disease process.

Although earlier studies on the epidemiology of Coats disease [3–18] have reported on the sex and age distributions, only a few have determined the prevalence and incidence of Coats disease in the general population [9,17]. Japan has been developing large medical databases which has resulted in more epidemiological studies. The Japan Medical Data Center (JMDC) claims database is one of these large-scale medical databases in Japan. It is a commercial database that is owned by JMDC, Inc (Tokyo, Japan) [19]. The database contains claims data from inpatient, outpatient, and dispensing services, as well as demographic information, service dates, diagnoses, procedures, and prescriptions for insured patients. It primarily covers employees and their dependents enrolled in health insurance associations of major companies in Japan. However, it does not include employees of small and medium-sized enterprises, public employees, self-employed individuals, part-time workers, unemployed persons, and people aged 75-years-of-age.

This database has the following features that make it useful for epidemiological studies. Patients are anonymized but personal identifiers can be used to track patients across medical facilities even if they are transferred or seen at multiple facilities. It includes information on outpatients as well as inpatients, and it includes insurance subscribers other than the patients. Because the database includes information on the number, age, and sex of all insurance subscribers, the demographics of this population is well defined. Therefore, the prevalence and incidence in the JMDC database can be easily determined by comparing the number of patients with Coats disease to the number of all insured persons.

The JMDC claims database has been used extensively in medical research in Japan in recent years [19]. In the field of ophthalmology, studies on various ocular disorders have been performed on this database [20–34]. We have determined the prevalence and incidence of Coats disease in the JMDC claims database.

## Methods

### Research design and data source

This was a retrospective study using the JMDC claims database. The data in this database have existed since January 2005, and the number of insurance subscribers

has increased yearly. There was a large increase in the total number of insurance subscribers between the early data and the more recent data. For this reason, this study used data from the most recent five fiscal years, 2019–2023, which were provided to us by JMDC, Inc.

The mean number of insured individuals was 9,859,371/year for the five-year period we studied. The total population of Japan as of October 2023, as reported by the Statistics Bureau, Ministry of Internal Affairs and Communications, a Japanese government agency, was approximately 124.35 million [35]. This indicates that the number of insured individuals in our study group was 7.9% of the total Japanese population. The male-female ratio was 55% male to 45% female, and the age distribution was 75% adults ≥20-years-of-age and 25% were children <20-years-of age in the JMDC database.

The procedures used in this study complied with the Declaration of Helsinki, and they were reviewed and approved by the Ethics Committee of Medical Research, University of Occupational and Environmental Health, Japan. The need for informed consent was waived because the JMDC claims database is anonymous and cannot be used to identify individuals.

## Patient selection

Patients whose standardized disease name on their receipts was "Coats disease" were selected from all the insured patients who had been examined at a medical institution during the five-year period from April 2019 to March 2024. The standardized disease name is a name that eliminates minor differences in its expressions and ensures a one-to-one correspondence between disease and disease name. This was done for efficient processing of the disease name information in the Japanese medical information system. A standardized disease name is assigned a unique identifier (seven-digit numeric code) for each disease to facilitate information processing on a computer. In addition, it was also assigned a classification code of the International Classification of Diseases, 10th Revision (ICD-10) to correspond to the international classification.

The patients' data were assigned a personal identifier, and the year of visit, sex, date of birth, date at diagnosis, and application of a specific treatment code (see below) were collected. The database was accessed for the purpose of this study during the period from 11/10/2024–06/11/2024.

## Calculation of prevalence and incidence

The prevalence and incidence of Coats disease in the JMDC database were calculated as the primary outcome. The following procedures were used to calculate the prevalence and incidence. First, cross-tabulations of the sex and age were performed for each year for the number of patients with Coats disease, the number of patients when Coats disease was first diagnosed, and the total number of insured individuals. Second, a cross-tabulation for each year was summated to calculate the five-year mean. The prevalence was calculated by dividing the mean number of Coats patients visiting medical institution by the mean of the total number of insured persons in the JMDC population and multiplying by 100,000. Similarly, the incidence was calculated by dividing the mean number of patients diagnosed with Coats disease for the first time by the mean total number of insured persons in the JMDC population and multiplying by 100,000. Sex-, age group– (children and adults), and overall prevalence and incidence rates were age-adjusted using the 2015 standard population of Japan provided by the Ministry of Health, Labour and Welfare. Then the 95% confidence intervals (95% CI) were calculated for each prevalence and incidence.

## Calculation of treatment rates

The treatment rate of patients with Coats disease was calculated as the third outcome measure. The following procedure was used to calculate the treatment rate. First, the treatment of Coats disease was divided into retinal photocoagulation or vitrectomy according to the K-codes specified in the Japanese Medical Fee Points. The K-codes in the Medical Fee Points

are codes for surgeries approved by the Japanese Ministry of Health, Labour and Welfare. The number of patients diagnosed with Coats disease for the first time and either received or did not receive these treatments were combined by sex and age for the five years. The treatment rate was then calculated by dividing the number of patients who received treatment by the total number of patients. In addition, cross-tabulation tables were made for sex and age, and a chi-square test was performed for each table.

## Statistical analysis

Data tabulation and statistical analysis were performed using Microsoft Excel 2021 and IBM SPSS Statistics 26. Statistical significance was set at $P < 0.05$.

## Results

### Annual trends in patient numbers

The number of patients visiting a medical institution/fiscal year increased yearly (Table 1). The proportion of male patients ranged from 76.4% to 78.8%, while the proportion of female patients ranged from 23.6 to 21.2%. The males outnumbered the females by 3.2X to 3.7X with small variations in different years. The proportion of children under 20-years-of-age ranged from 39.2 to 44.2%, while the percentage of adults over 20-years-of-age ranged from 60.8 to 55.8%. Adults were 1.3 to 1.6 times more likely to be examined than children, and more adults than children were examined in any given year.

The number of patients at diagnosis/fiscal year has decreased since 2021 compared to 2019 and 2020 (Table 2). The proportion of male patients ranged from 65.4% to 86.7%, while the proportion of female patients ranged from 34.6% to 13.3%. The male-to-female ratio ranged from 1.9 to 6.5 and showed greater year-to-year variation than the number of patients visiting medical institutions. The proportion of children under 20-years-of-age ranged from 24.4% to 46.1%, while that of adults over 20-years-of-age ranged from 75.6% to 53.9%. The proportion of adults was 1.2 to 3.1 times higher than that of children, with substantial year-to-year fluctuations.

### Prevalence

The crude prevalence of Coats disease in the JMDC database was calculated as the mean number of Coats patients seen during the five-year period and the total number of insured individuals during the five-year study period. The age-adjusted prevalence of Coats disease was 1.7/100,000 for the total five-year period (95% CI: 1.4–1.9, Table 3), and it was

**Table 1. Annual trends in the number of patients visiting medical institutions for Coats disease.**

|  | 2019* | 2020 | 2021 | 2022 | 2023 | Mean | Median |
|---|---|---|---|---|---|---|---|
| Total | 168 | 198 | 208 | 232 | 250 | 211.2 | 208 |
| Sex |  |  |  |  |  |  |  |
| Male | 129 (76.8)** | 152 (76.8) | 159 (76.4) | 181 (78.0) | 197 (78.8) | 163.6 (77.5) | 159 (76.4) |
| Female | 39 (23.2) | 46 (23.2) | 49 (23.6) | 51 (22.0) | 53 (21.2) | 47.6 (22.5) | 49 (23.6) |
| Age |  |  |  |  |  |  |  |
| Under 20 | 71 (42.3) | 82 (41.4) | 92 (44.2) | 95 (40.9) | 98 (39.2) | 87.6 (41.5) | 92 (44.2) |
| Over 20 | 97 (57.7) | 116 (58.6) | 116 (55.8) | 137 (59.1) | 152 (60.8) | 123.6 (58.5) | 116 (55.8) |

*Year is the Japanese fiscal year (April to March of the following year).

**The values in parentheses indicate percentages.

**Table 2. Annual trends in the number of patients newly diagnosed with Coats disease.**

|  | 2019* | 2020 | 2021 | 2022 | 2023 | Mean | Median |
|---|---|---|---|---|---|---|---|
| Total | 115 | 128 | 78 | 67 | 45 | 86.6 | 78 |
| Sex |  |  |  |  |  |  |  |
| Male | 88 (76.5)** | 103 (80.5) | 51 (65.4) | 49 (73.1) | 39 (86.7) | 66 (76.2) | 51 (65.4) |
| Female | 27 (23.5) | 25 (19.5) | 27 (34.6) | 18 (26.9) | 6 (13.3) | 20.6 (23.8) | 25 (32.1) |
| Age |  |  |  |  |  |  |  |
| Under 20 | 53 (46.1) | 49 (38.3) | 28 (35.9) | 23 (34.3) | 11 (24.4) | 32.8 (37.9) | 28 (35.9) |
| Over 20 | 62 (53.9) | 79 (61.7) | 50 (64.1) | 44 (65.7) | 34 (75.6) | 53.8 (62.1) | 50 (64.1) |

*Year numbers indicate Japanese fiscal year (April to March of the following year).

**The values in parentheses indicate percentages.

**Table 3. Prevalence of Coats disease.**

|  | Male | Female | Total |
|---|---|---|---|
| Age-adjusted rate (Total) | 2.4 (2.0-2.8)* | 0.9 (0.6-1.2) | 1.7 (1.4-1.9) |
| Crude rate (Total) | 3.0 (2.6-3.5) | 1.1 (0.8-1.4) | 2.1 (1.9-2.4) |
| Age |  |  |  |
| Under10 | 6.4 (4.3-8.5) | 1.4 (0.4-2.3) | 3.9 (2.8-5.1) |
| 10-19 | 5.0 (3.3-6.7) | 1.6 (0.6-2.6) | 3.4 (2.4-4.3) |
| 20-29 | 3.4 (2.1-4.7) | 0.9 (0.1-1.6) | 2.3 (1.5-3.1) |
| 30-39 | 2.4 (1.3-3.4) | 0.7 (0.1-1.4) | 1.7 (1.0-2.3) |
| 40-49 | 2.3 (1.3-3.2) | 1.1 (0.4-1.8) | 1.7 (1.1-2.3) |
| 50-59 | 1.7 (0.9-2.5) | 0.8 (0.2-1.4) | 1.3 (0.8-1.8) |
| Over 60 | 1.3 (0.3-2.1) | 1.1 (0.0-2.2) | 1.2 (0.5-2.0) |
|  |  |  |  |
| Under 20** | 5.7 (4.3-7.0) | 1.5 (0.8-2.2) | 3.6 (2.9-4.4) |
| Over 20** | 1.7 (1.3-2.1) | 0.7 (0.4-1.0) | 1.3 (1.0-1.5) |

*The values in parentheses indicate the 95% confidence interval.

**The prevalence rates were age-adjusted.

2.4/100,000 (95% CI: 2.0–2.8) for males and 0.9/100,000 (95% CI: 0.6–1.2) for females. The age-adjusted prevalence for children <20 years was 3.6/100,000 (95% CI: 2.9–4.4), and that for adults ≥20 years was 1.3/100,000 (95% CI: 1.0–1.5) in the JMDC database. For children, the prevalence was 5.7/100,000 (95% CI: 4.3–7.0) for boys and 1.5/100,000 (95% CI: 0.8–2.2) for girls. For adults, the prevalence was 1.7/100,000 (95% CI: 1.3–2.1) for men and 0.7/100,000 (95% CI: 0.4–1.0) for women.

The age-adjusted prevalence of Coats disease in 2019 was 1.5/100,000 (95% CI: 1.3–1.8), but in 2023, it had increased to 1.8/100,000 (95% CI: 1.6–2.1). Examination of the tables shows that there was a steady increase in the number of cases of Coats disease yearly (Table 4). When separated by the sex and age, an increasing trend in the prevalence was observed for all groups including men, children, and adults but not women.

### Incidence

The crude incidence of Coats disease in Japan was calculated based on the five-year mean number of Coats patients at the initial diagnosis and the total number of insured individuals. The age-adjusted incidence was 0.7/100,000 (95% CI: 0.5–0.9; Table 5). The age-adjusted incidence was 1.0/100,000 for males (95% CI: 0.7–1.3) and 0.4/100,000 for females (95% CI: 0.2–0.6).

Similar to the prevalence, the age-adjusted incidence was calculated for children under 20-years-of-age and adults aged ≥20 years-of-age. The age-adjusted incidence for children was 1.4/100,000 (95% CI: 0.9–1.8), and for adults, it was 0.6/100,000 (95% CI: 0.4–0.7). For the children, the incidence was 2.1/100,000 for boys (95% CI: 1.3–2.9) and 0.6/100,000 for girls (95% CI: 0.2–1.0). For adults, the incidence was 0.8/100,000 for men (95% CI: 0.5–1.0) and 0.3/100,000 for women (95% CI: 0.1–0.5).

The annual change in the age-adjusted incidence was 1.1/100,000 for 2019 and 2020, but it continued to decline after 2021, reaching 0.4/100,000 in 2023 (Table 6). When separated by sex, the age-adjusted incidence for males was 1.5/100,000 in 2019 (95% CI: 1.2–1.8) and 0.5/100,000 in 2023 (95% CI: 0.3–0.7). The age-adjusted incidence for females decreased from 0.6/100,000 in 2019 (95% CI: 0.3–0.8) to 0.2/100,000 in the 2023 (95% CI: 0.0–0.3). The age-adjusted incidence for children was 2.4/100,000 (95% CI: 1.8–3.0) in 2019 compared to 0.4/100,000 (95% CI: 0.2–0.7) in 2023. The age-adjusted incidence among adults decreased from 0.8/100,000 (95% CI: 0.6–1.0) in 2019 to 0.3/100,000 (95% CI: 0.2–0.5) in 2023, indicating a decreasing trend.

**Table 4. Annual trends in the age-adjusted prevalence of Coats disease.**

|  | 2019* | 2020 | 2021 | 2022 | 2023 |
|---|---|---|---|---|---|
| Total | 1.5 (1.3-1.8)** | 1.6 (1.4-1.9) | 1.6 (1.4-1.9) | 1.7 (1.5-2.0) | 1.8 (1.6-2.1) |
| Sex |  |  |  |  |  |
| Male | 2.1 (1.7-2.5) | 2.3 (1.9-2.7) | 2.3 (1.9-2.7) | 2.5 (2.0-2.9) | 2.6 (2.2-3.1) |
| Female | 0.9 (0.6-1.2) | 0.8 (0.6-1.1) | 0.8 (0.6-1.1) | 0.9 (0.6-1.2) | 0.9 (0.6-1.2) |
| Age |  |  |  |  |  |
| Under20 | 3.2 (2.5-3.9) | 3.5 (2.7-4.2) | 3.8 (3.0-4.6) | 3.8 (3.0-4.5) | 3.8 (3.1-4.6) |
| Over 20 | 1.2 (0.9-1.4) | 1.2 (1.0-1.5) | 1.2 (0.9-1.4) | 1.3 (1.0-1.6) | 1.4 (1.1-1.7) |

*Year numbers indicate Japanese fiscal year (April to March of the following year).

**The values in parentheses indicate the 95% confidence interval.

**Table 5. Incidence of Coats disease.**

|  | Male | Female | Total |
|---|---|---|---|
| Age-adjusted rate (Total) | 1.0 (0.7-1.3)* | 0.4 (0.2-0.6) | 0.7 (0.5-0.9) |
| Crude rate (Total) | 1.2 (0.9-1.5) | 0.5 (0.3-0.7) | 0.9 (0.7-1.1) |
| Age |  |  |  |
| Under10 | 1.9 (0.8-3.1) | 0.6 (0.1-1.3) | 1.3 (0.6-2.0) |
| 10-19 | 2.2 (1.1-3.3) | 0.6 (0.0-1.2) | 1.4 (0.8-2.1) |
| 20-29 | 1.5 (0.6-2.3) | 0.4 (0.1-0.9) | 1.0 (0.5-1.5) |
| 30-39 | 0.9 (0.3-1.6) | 0.3 (0.1-0.7) | 0.7 (0.3-1.1) |
| 40-49 | 0.6 (0.1-1.0) | 0.4 (0.0-0.9) | 0.5 (0.2-0.8) |
| 50-59 | 1.0 (0.4-1.7) | 0.5 (0.0-0.9) | 0.8 (0.4-1.2) |
| Over 60 | 0.9 (0.1-1.7) | 0.5 (0.0-1.2) | 0.7 (0.1-1.3) |
|  |  |  |  |
| Under20** | 2.1 (1.3-2.9) | 0.6 (0.2-1.0) | 1.4 (0.9-1.8) |
| Over 20** | 0.8 (0.5-1.0) | 0.3 (0.1-0.5) | 0.6 (0.4-0.7) |

*The values in parentheses indicate the 95% confidence interval.

**The incidence rates were age-adjusted.

**Table 6. Annual trends in the age-adjusted incidence of Coats disease.**

|  | 2019* | 2020 | 2021 | 2022 | 2023 |
|---|---|---|---|---|---|
| Total | 1.1 (0.9-1.3)** | 1.1 (0.9-1.3) | 0.6 (0.5-0.8) | 0.5 (0.4-0.7) | 0.4 (0.2-0.5) |
| Sex |  |  |  |  |  |
| Male | 1.5 (1.2-1.8) | 1.7 (1.4-2.1) | 0.8 (0.5-1.0) | 0.7 (0.4-0.9) | 0.5 (0.3-0.7) |
| Female | 0.6 (0.3-0.8) | 0.4 (0.2-0.6) | 0.5 (0.3-0.7) | 0.4 (0.2-0.5) | 0.2 (0.0-0.3) |
| Age |  |  |  |  |  |
| Under20 | 2.4 (1.8-3.0) | 2.1 (1.5-2.6) | 1.2 (0.7-1.6) | 0.9 (0.5-1.3) | 0.4 (0.2-0.7) |
| Over 20 | 0.8 (0.6-1.0) | 0.9 (0.7-1.1) | 0.5 (0.4-0.7) | 0.4 (0.3-0.6) | 0.3 (0.2-0.5) |

*Year numbers indicate Japanese fiscal year (April to March of the following year).

**The values in parentheses indicate the 95% confidence interval.

## Treatment Rates

Over the five-year examination period, 95 patients with Coats disease received treatment, and 338 patients did not receive treatment. Thus, the overall treatment rate was 21.9%. According to the cross-tabulation table of treatment status and age, the treatment rate was 28.0% for children <20-years-of-age and 18.2% for adults aged ≥20-years (Table 7). The results of the chi-square test showed that there was a significant difference in the cross-tabulation table of treatment status and age, with children having a significantly higher treatment rates than adults ($\chi^2$ (1) = 5.75, $P$ = 0.016). The cross-tabulation table of treatment status and sex showed that the treatment rate was 23.0% for males and 18.4% for females (Table 8). The results of the chi-square test showed no statistically significant difference in the cross-tabulation table of treatment status and sex.

## Discussion

The age-adjusted prevalence of Coats disease in the JMDC claims database was 1.7/100,000, and the age-adjusted incidence was 0.7/100,000 based on the five-year membership numbers. Morris et al. (2010) reported an incidence of 0.09/100,000 in the United Kingdom (UK) [9] and Dorji et al. (2023) reported a prevalence of 25.3/100,000 in India [17]. Morris et al. (2010) examined 55 cases from baseline questionnaires returned by ophthalmologists who reported 72 new cases of Coats disease voluntarily reported by ophthalmologists through the British Ophthalmological Surveillance Unit (BOSU) in 2008 [9]. Dorji et al. (2023) extracted 675 patients diagnosed with Coats disease from electronic medical records of 2,664,906 new patients who visited an ophthalmology network in India between 2010 and 2021. Using the 2.66 million new patients as the study population, they reported that the prevalence [17]. This study found an incidence about eight times higher than the UK study and a prevalence about one-fifteenth of the Indian study. Three factors may explain these discrepancies: (1) completeness of patients' data, (2) differences in study population size, and (3) racial differences.

Regarding point (1), Morris et al. (2010) calculated an incidence of 0.09/100,000 as a minimum estimate rather than a definitive value [9]. This estimate may be low due to the reliance on voluntary ophthalmologist reports and possibly missing cases. Dorji et al. (2023) used electronic medical records which likely examined most patients [17]. We utilized a claims database encompassing the majority of diagnosed cases.

With reference to point (2), The UK study used the entire national population (61.4 million) [9], the Indian study used new ophthalmology patients (2.66 million) [17], and our study used 9.86 million insured individuals. If the number of patients remains unchanged while the population size increases, the estimated rate would decrease. Therefore, the population size and resulting estimates in this study are intermediate between those of the two previous studies. In the UK study [9], Coats disease cases were likely undercounted due to reliance on voluntary reports from ophthalmologists, making the total UK population an unsuitable study base. The Indian study [17] used newly presenting ophthalmology patients

**Table 7. Cross-tabulation of treatment status and age.**

| Age | No treatment | Treatment | Total | % |
|---|---|---|---|---|
| Under 20 | 118 | 46 | 164 | 28.0 |
| Over 20 | 220 | 49 | 269 | 18.2 |
| Total | 338 | 95 | 433 | 21.9 |

**Table 8. Cross-tabulation of treatment status and sex.**

| Sex | No treatment | Treatment | Total | % |
|---|---|---|---|---|
| Male | 254 | 76 | 330 | 23.0 |
| Female | 84 | 19 | 103 | 18.4 |
| Total | 338 | 95 | 433 | 21.9 |

as the study population; therefore, the reported prevalence reflects rates among eye care patients and likely overestimates the prevalence in the general population.

Our study's population comprised all insurance subscribers in the database that represented about 7.9% (approximately 1/12) of the total Japanese population. Whether this group reflects the demographics of the national population can be assessed by examining potential biases. The age distribution of the study population was adjusted to the age structure of the standard Japanese population. On the other hand, as the JMDC database included mainly high-income employees from large companies, a socioeconomic bias exists. However, since Coats disease primarily affects infants and is unlikely influenced by family income, this bias likely has minimal impact on prevalence or incidence estimates.

Regarding point (3), the ethnicity of this and previous studies differ. Japan is predominantly Mongoloid, the UK predominantly Caucasian, and India includes a mixture of ethnic groups. While Coats disease is thought to lack an ethnic bias [5], the evidence is limited.

The age-adjusted prevalence of Coats disease was 2.4/100,000 in males and 0.9/100,000 females, making it approximately 2.7 times more common in males. Similarly, the age-adjusted incidence was 1.0/100,000 in males and 0.4/100,000 in females, indicating a 2.5-fold higher rate in males. These findings are consistent with previous studies conducted in other countries [3,5,8–18]. This study clearly confirmed that Coats disease is more prevalent in males.

Children under 20 had an age-adjusted prevalence of 3.6/100,000 and an age-adjusted incidence of 1.4/100,000, while adults had an age-adjusted prevalence of 1.3/100,000 and an age-adjusted incidence of 0.6/100,000. Although Coats disease is typically more common in children, this study found a higher adult proportion than previous reports. In adults, the disease usually presents with localized lesions and is slowly progressive [7] with severe cases more common in children and mild cases in adults [4,6,18]. A higher rate of adult-onset cases has been reported in Korea compared to non-Asian populations [13]. Some adult cases may have been undiagnosed in childhood due to mild symptoms and slow disease progression.

In relation to sex and age, the prevalence ratio between males and females was 1:0.26 in children (5.7 vs. 1.5) and 1:0.45 in adults (1.7 vs. 0.7). Similarly, the incidence ratio was 1:0.29 in children (2.1 vs. 0.6) and 1:0.44 in adults (0.8 vs. 0.3). These findings indicate that the proportion of female cases, in both prevalence and incidence, was higher in adults than in children. A study from Taiwan [8] reported that the proportion of female Coats patients was higher among those aged ≥20 years compared to those under 20. However, studies focusing on the characteristics of female patients with Coats disease are limited, and consistent findings have yet to be reported. While one study [12] reported a lower proportion of male patients in advanced-stage cases, other studies [15,16] found no sex differences in age at diagnosis or disease severity. These inconsistencies may, in part, be due to the small number of female patients included in the studies.

The annual trend in prevalence of Coats disease showed a gradual increase in the number of patients, whereas the incidence has continued to decline since fiscal year 2021. The increase in prevalence (Table 4) reflects the increase in the number of patients visiting medical institutions (Table 1). In contrast, the decline in incidence is likely due to the impact of the COVID-19 pandemic which led to reduced visits to medical facilities. While the number of new patients decreased during the pandemic, those who had already visited a medical institution by fiscal year 2020 continued to do so resulting in a discrepancy between the annual trends of prevalence and incidence.

The treatment rate at the time of the diagnosis of Coats disease was 22%, and children had higher treatment rates compared to adults. This finding is consistent with previous studies indicating that severe cases are more common in children than in adults [4,6,18]. The lower treatment rate observed in adults may be due to a higher proportion of mild cases that did not require retinal photocoagulation or vitrectomy.

There are limitations in this study. First, large-scale medical databases typically employ a standardized formats across all medical departments, which do not include ophthalmology-specific evaluations. As a result, it is not possible to investigate ophthalmology specific epidemiological characteristics such as laterality and visual acuity that have been examined in previous studies. Second, many previous studies have assessed the disease severity using Shields' staging

classification [2] and have examined its association with other attributes. In our study, due to the high proportion of adult cases, we assumed the relationship between treatment rates and severity indirectly. However, it is difficult to directly analyze the association between severity and other attributes. Third, in this study, patients were identified solely based on the standardized disease name "Coats disease." Because Coats disease is typically diagnosed in childhood and presents with characteristic fundus findings, the diagnostic coding accuracy is considered to be high. However, as the database contains only coded information, it is difficult to verify potential misclassification for each case as we did in our own clinical cases. Therefore, the possibility of case misclassification cannot be completely excluded in this study. Fourth, the treatment options selected here were retinal photocoagulation and vitrectomy. However, adjunctive therapies such as intravitreal injection of anti-VEGF agents are also currently used as off-label treatments [36]. Therefore, the treatment rate in this study reflects the proportion of standard interventions and does not include all treatment modalities.

In conclusion, this study used the JMDC claims database to estimate the prevalence and incidence of Coats disease in Japan. The age-adjusted prevalence was 1.7/100,000 and the age-adjusted incidence was 0.7/100,000 falling between prior estimates from the UK and India. The prevalence and incidence were higher in males than in females, and in children than in adults. The proportion of females was higher in adults than in children.

## Author contributions

**Conceptualization:** Nobuhisa Ochiai, Hiroyuki Kondo.

**Data curation:** Nobuhisa Ochiai, Kenji Fujimoto.

**Formal analysis:** Nobuhisa Ochiai, Kenji Fujimoto.

**Funding acquisition:** Nobuhisa Ochiai.

**Investigation:** Nobuhisa Ochiai.

**Methodology:** Nobuhisa Ochiai, Kenji Fujimoto, Kazuma Oku, Hiroyuki Kondo.

**Resources:** Nobuhisa Ochiai, Kenji Fujimoto.

**Supervision:** Hiroyuki Kondo.

**Writing – original draft:** Nobuhisa Ochiai, Hiroyuki Kondo.

**Writing – review & editing:** Nobuhisa Ochiai, Kenji Fujimoto, Kazuma Oku, Hiroyuki Kondo.

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
