## [Decision Letter · Decision Letter 0]

20 Oct 2025

Dear Dr. KONDO,

Thank you for submitting your manuscript to PLOS ONE. After careful consideration, we feel that it has merit but does not fully meet PLOS ONE’s publication criteria as it currently stands. Therefore, we invite you to submit a revised version of the manuscript that addresses the points raised during the review process.

We look forward to receiving your revised manuscript.

Kind regards,

Tatsuya Inoue

Academic Editor

PLOS ONE

Journal Requirements:

“This work was supported by JSPS KAKENHI Grant Number 24K12756.”

“NO received JSPS KAKENHI (grant number 24K12756). The Japan Society for the Promotion of Science (JSPS). http://www.jsps.go.jp/english/e-grants/index.html. The funders had no role in study design, data collection and analysis, decision to publish, or preparation of the manuscript.”

Reviewers' comments:

Reviewer's Responses to Questions

**Comments to the Author**

1. Is the manuscript technically sound, and do the data support the conclusions?

Reviewer #1: Yes

Reviewer #2: Yes

2. Has the statistical analysis been performed appropriately and rigorously?

Reviewer #1: Yes

Reviewer #2: Yes

3. Have the authors made all data underlying the findings in their manuscript fully available?

Reviewer #1: Yes

Reviewer #2: Yes

4. Is the manuscript presented in an intelligible fashion and written in standard English?

Reviewer #1: Yes

Reviewer #2: Yes

Reviewer #1: This is the first large-scale epidemiologic study of Coats disease conducted in Japan, making it a valuable contribution to the literature on this rare retinal disorder. The use of the JMDC Claims database allows for age- and sex-stratified analyses as well as treatment rate calculations, which add important clinical context. The finding that prevalence and incidence in children are approximately twice those in adults is noteworthy and aligns with known clinical patterns.

Coats disease is predominantly diagnosed in children and has distinctive fundus findings, which likely contribute to high diagnostic coding accuracy. This suggests that the disease is well-suited for claims-based epidemiologic studies, with a relatively low risk of case misclassification.

However, some revisions are necessary regarding the following points.

In the ≥20-year-old group in particular, the study population is biased because the JMDC database primarily includes employees enrolled in corporate health insurance and their dependents. This results in underrepresentation of elderly individuals (especially those ≥75 years), self-employed persons, and the unemployed. As the age, occupational, and socioeconomic distributions differ from those of the general population, adjustments such as age standardization are recommended to produce more accurate, population-based estimates.

Reviewer #2: This is a valuable study that addresses a significant gap in the epidemiological data for Coats disease in Japan. Overall, the manuscript provides important insights and is worthy of publication. However, several points should be addressed to improve the clarity and depth of the paper. I recommend Minor Revision before acceptance.

Minor points;

1. Absence of Age-Adjusted Prevalence:

While the study presents age-stratified prevalence rates, it does not provide an age-adjusted prevalence rate standardized to a reference population (e.g., the general Japanese population). The age structure of the JMDC database (75% adults, 25% children) likely differs from that of the general population in Japan. Therefore, the calculated crude prevalence rate (2.1/100,000) may be influenced by the specific age demographics of the database. Presenting an age-adjusted rate would enhance the generalizability of the findings and allow for more accurate comparisons with other populations.

2. Limitations of Case Definition:

The study identifies patients based solely on the standardized diagnostic name "Coats disease". The authors should more explicitly discuss the limitations of this approach in the Discussion section. There is a potential for misclassification. Acknowledging limited diagnostic accuracy, without validation analysis, is a key limitation of the study.

3. Definition of Treatment Rate:

The definition of treatment is restricted to retinal photocoagulation and vitrectomy. While these are primary interventions, adjunctive therapies such as intravitreal anti-VEGF injections are now common. It should be clarified in the limitations that the reported "treatment rate" specifically reflects the rate of major surgical interventions and may not capture all therapeutic modalities.

**Do you want your identity to be public for this peer review?** For information about this choice, including consent withdrawal, please see our Privacy Policy

Reviewer #1: No

Reviewer #2: No

---

## [Author Response · Author response to Decision Letter 1]

12 Nov 2025

Thank you for taking the time to review our manuscript and for providing thoughtful comments. We have revised our manuscript by incorporating all comments.

Editor:

The grant information has been removed from the Acknowledgements Section. The following sentence has been added the on-line statement part: “JSPS KAKENHI (grant number 24K12756) for NO”.

Regarding the Public Sharing of Raw Data (the dataset used for analysis in this study):

The data used in this study were obtained from the JMDC Claims Database under a contractual agreement between JMDC Inc. and the University of Occupational and Environmental Health, Japan. Because the terms of this agreement prohibit the authors from sharing the data with third parties, the dataset cannot be made publicly available.

However, as the dataset is commercially available for purchase from JMDC Inc., the authors did not have any special access privileges that others would not have. Other researchers may access the same dataset directly from JMDC Inc. by paying the required usage fee. For inquiries regarding data access, please contact JMDC Inc. through their website:

(Japanese) https://www.jmdc.co.jp/inquiry/

(English) https://www.jmdc.co.jp/en/inquiry/

Reviewer #1:

In the ≥20-year-old group in particular, the study population is biased because the JMDC database primarily includes employees enrolled in corporate health insurance and their dependents. This results in underrepresentation of elderly individuals (especially those ≥75 years), self-employed persons, and the unemployed. As the age, occupational, and socioeconomic distributions differ from those of the general population, adjustments such as age standardization are recommended to produce more accurate, population-based estimates.

[Answer] Thank you for the valuable comment. We have changed the prevalence and incident data to age-adjusted data in Abstract, text and Tables 3-6. Accordingly, the detail of the adjustments was added in the Methods section: “Sex-, age group– (children and adults), and overall prevalence and incidence rates were age-adjusted using the 2015 standard population of Japan provided by the Ministry of Health, Labour and Welfare”. Additionally, we have added the following sentence to lines 300–301 of the paragraph discussing bias in the study population: “The age distribution of the study population was adjusted to the age structure of the standard Japanese population.”

Reviewer #2:

Minor points;

1. Absence of Age-Adjusted Prevalence:

While the study presents age-stratified prevalence rates, it does not provide an age-adjusted prevalence rate standardized to a reference population (e.g., the general Japanese population). The age structure of the JMDC database (75% adults, 25% children) likely differs from that of the general population in Japan. Therefore, the calculated crude prevalence rate (2.1/100,000) may be influenced by the specific age demographics of the database. Presenting an age-adjusted rate would enhance the generalizability of the findings and allow for more accurate comparisons with other populations.

[Answer] Thank you for the valuable comment. We have changed the prevalence and incident data to age-adjusted data in Abstract, text and Tables 3-6. Accordingly, the detail of the adjustments was added in the Methods section: “Sex-, age group– (children and adults), and overall prevalence and incidence rates were age-adjusted using the 2015 standard population of Japan provided by the Ministry of Health, Labour and Welfare”. Additionally, we have added the following sentence to lines 300–301 of the paragraph discussing bias in the study population: “The age distribution of the study population was adjusted to the age structure of the standard Japanese population.”

2. Limitations of Case Definition:

The study identifies patients based solely on the standardized diagnostic name "Coats disease". The authors should more explicitly discuss the limitations of this approach in the Discussion section. There is a potential for misclassification. Acknowledging limited diagnostic accuracy, without validation analysis, is a key limitation of the study.

[Answer] Thank you for the valuable suggestion. The following statement has been added in the limitation section:

“Third, in this study, patients were identified solely based on the standardized disease name “Coats’ disease.” Because Coats’ disease is typically diagnosed in childhood and presents with characteristic fundus findings, the diagnostic coding accuracy is considered to be high. However, as the database contains only coded information, it is difficult to verify potential misclassification for each case as we did in our own clinical cases. Therefore, the possibility of case misclassification cannot be completely excluded in this study.”

3. Definition of Treatment Rate:

The definition of treatment is restricted to retinal photocoagulation and vitrectomy. While these are primary interventions, adjunctive therapies such as intravitreal anti-VEGF injections are now common. It should be clarified in the limitations that the reported "treatment rate" specifically reflects the rate of major surgical interventions and may not capture all therapeutic modalities.

[Answer] Thank you for the valuable suggestion. The following statement has been added in the limitation section as well: “Forth, the treatment options selected here were retinal photocoagulation and vitrectomy. However, adjunctive therapies such as intravitreal injection of anti-VEGF agents are also currently used as off-label treatments [36]. Therefore, the treatment rate in this study reflects the proportion of standard interventions and does not include all treatment modalities.”

---

## [Decision Letter · Decision Letter 1]

10 Dec 2025

Prevalence and incidence of Coats disease in a large claims database

PONE-D-25-38385R1

Dear Dr. KONDO,

We’re pleased to inform you that your manuscript has been judged scientifically suitable for publication and will be formally accepted for publication once it meets all outstanding technical requirements.

Kind regards,

Tatsuya Inoue

Academic Editor

PLOS One

Additional Editor Comments (optional):

Reviewers' comments:

Reviewer's Responses to Questions

**Comments to the Author**

Reviewer #1: All comments have been addressed

Reviewer #2: All comments have been addressed

2. Is the manuscript technically sound, and do the data support the conclusions?

Reviewer #1: Yes

Reviewer #2: Yes

3. Has the statistical analysis been performed appropriately and rigorously?

Reviewer #1: Yes

Reviewer #2: Yes

4. Have the authors made all data underlying the findings in their manuscript fully available?

Reviewer #1: Yes

Reviewer #2: Yes

5. Is the manuscript presented in an intelligible fashion and written in standard English?

Reviewer #1: Yes

Reviewer #2: Yes

Reviewer #1: The authors have appropriately addressed the comments I raised in my review, and I am pleased to recommend this manuscript for acceptance.

Reviewer #2: Typo; (p21 l367) "Forth"->"Fourth."

**Do you want your identity to be public for this peer review?** For information about this choice, including consent withdrawal, please see our Privacy Policy

Reviewer #1: **Yes: ** SHIN TANAKA

Reviewer #2: No

---

## [Editor Report · Acceptance letter]

PONE-D-25-38385R1

PLOS One

Dear Dr. Kondo,

I'm pleased to inform you that your manuscript has been deemed suitable for publication in PLOS One. Congratulations! Your manuscript is now being handed over to our production team.

Kind regards,

on behalf of

Dr. Tatsuya Inoue

Academic Editor

PLOS One